# Diagnostic Accuracy of AI for Opportunistic Screening of Abdominal Aortic Aneurysm in CT: A Systematic Review and Narrative Synthesis

**DOI:** 10.3390/diagnostics12123197

**Published:** 2022-12-16

**Authors:** Maria R. Kodenko, Yuriy A. Vasilev, Anton V. Vladzymyrskyy, Olga V. Omelyanskaya, Denis V. Leonov, Ivan A. Blokhin, Vladimir P. Novik, Nicholas S. Kulberg, Andrey V. Samorodov, Olesya A. Mokienko, Roman V. Reshetnikov

**Affiliations:** 1Research and Practical Clinical Center for Diagnostics and Telemedicine Technologies of the Moscow Health Care Department, Petrovka Street, 24, Building 1, 127051 Moscow, Russia; 2Department of Biomedical Technologies, Bauman Moscow State Technical University, 2nd Baumanskaya Street, 5, Building 1, 105005 Moscow, Russia; 3Department of Information and Internet Technologies, I.M. Sechenov First Moscow State Medical University (Sechenov University), Trubetskaya Street, 8, Building 2, 119991 Moscow, Russia; 4Department of Fundamentals of Radio Engineering, Moscow Power Engineering Institute, Krasnokazarmennaya Street, 14, Building 1, 111250 Moscow, Russia; 5Federal Research Center “Computer Science and Control” of Russian Academy of Sciences, Vavilova Street, 44, Building 2, 119333 Moscow, Russia

**Keywords:** abdominal aortic aneurysm, opportunistic screening, computed tomography, artificial intelligence, QUADAS

## Abstract

In this review, we focused on the applicability of artificial intelligence (AI) for opportunistic abdominal aortic aneurysm (AAA) detection in computed tomography (CT). We used the academic search system PubMed as the primary source for the literature search and Google Scholar as a supplementary source of evidence. We searched through 2 February 2022. All studies on automated AAA detection or segmentation in noncontrast abdominal CT were included. For bias assessment, we developed and used an adapted version of the QUADAS-2 checklist. We included eight studies with 355 cases, of which 273 (77%) contained AAA. The highest risk of bias and level of applicability concerns were observed for the “patient selection” domain, due to the 100% pathology rate in the majority (75%) of the studies. The mean sensitivity value was 95% (95% CI 100–87%), the mean specificity value was 96.6% (95% CI 100–75.7%), and the mean accuracy value was 95.2% (95% CI 100–54.5%). Half of the included studies performed diagnostic accuracy estimation, with only one study having data on all diagnostic accuracy metrics. Therefore, we conducted a narrative synthesis. Our findings indicate high study heterogeneity, requiring further research with balanced noncontrast CT datasets and adherence to reporting standards in order to validate the high sensitivity value obtained.

## 1. Introduction

Abdominal aortic aneurysm (AAA) has no specific symptoms and can be asymptomatic at the early stages [1]. When untreated, AAA can lead to an aortic rupture, a life-threatening condition with an overall mortality of 80% [2,3,4]. Presently, the accepted diagnostic modality for AAA screening is ultrasonic imaging, while computed tomography angiography (CTA) remains the “gold standard” for treatment planning [2]. Compared to ultrasonography, the advantages of CT include superior image quality, a lower operator dependency, three-dimensional reconstruction, and the possibility of a retrospective data audit [2]. A CT is also more sensitive to aortic dilation than ultrasonography [5]. The radiation exposure associated with CT restricts its application as a screening method, but CT data can be used for opportunistic AAA detection either while reporting the study or via retrospective analysis of scans with the abdominal aorta in the field of view. According to the results of such audits, non-reported AAAs ranged from 0.4% (one of 261 patients) [6] to 5.8% (187 of 3246 patients) [7]. Taking into account the high volume of accumulated CT data (for example, in the USA the number of CT examinations was 278.5 per 1000 inhabitants in 2019 [8]), opportunistic screening could yield an increase in early diagnosed aneurysms in the population. Despite its potential, opportunistic screening for AAA at the CT exam remains a challenging task. The reported radiologist diagnostic accuracy for this task depends on the aneurysm’s size, with the lowest sensitivity of 0.52 for the small ones (30–39 mm) [7]. The radiologists’ errors consist of false-negatives and incorrect classification due to human-based or technical reasons [9]. AAA detection is also complicated by the measurement ambiguity of the key diagnostic parameter, the aneurysmal sac maximum transverse diameter [2].

Artificial intelligence (AI) has already shown its high potential for CT image-processing automatization [5,10,11] and promises to be a powerful assistant for radiologists’ practice. Automatization of diagnostic information processing has several advantages. First, it provides a tool for a retrospective audit of big data. Second, AI yields reproducible and precise measurements, addressing the ambiguity issue of human experts.

The aim of this review was to quantify the diagnostic accuracy of AI algorithms for AAA detection by noncontrast CT, regardless of the aneurysm’s size.

## 2. Materials and Methods

This systematic review was planned, conducted, and reported in accordance with the PRISMA statement [12], and the full protocol was registered on PROSPERO on 25 July 2021, before the literature search (PROSPERO ID CRD42021264021). The target condition evaluated was abdominal aortic aneurysm, defined as a permanent localized pathological dilatation of the abdominal aorta with a diameter greater than 3 cm or more than 50% larger than the nondilated part [13]. We also defined a negative diagnosis for AAA as the absence of abdominal aortic dilatation corresponding to the criteria above. In this review, we focused on the opportunistic screening model—the interpretation of noncontrast CT studies. For that reason, we considered CT studies without intravenous contrast and containing the aorta abdominal region in the field of view.

The index test was AAA detection by an AI algorithm. AI should have provided enough information to conclude whether an AAA was present or absent from the noncontrast CT images of the abdominal area. The image processing could be of any AI type, but the segmentation should have been fully automatic.

Manual expert segmentation was considered to be the reference standard, or “ground truth” (GT). The quality of the reference standard was estimated either by the level of expertise for a single observer or by any of the agreement metrics [14].

### 2.1. Search Methods for the Identification of Studies

The PubMed database was used as the main data source. Additional data (including gray literature) were searched using the Google Scholar search engine. The last search date was 2 February 2022. As sources of grey literature, we explored commercial websites with AI solutions for automated aortic segmentation in noncontrast CTs, because they rarely report results in articles [15]. The key concepts were the following: artificial intelligence, CT, abdominal aortic aneurysm, and opportunistic screening. We defined the most important and specific components of the query following the method proposed by Bramer et al. [16]. The terms included medical object (AAA), technical subject (AI), and type of intervention (detection or segmentation). Despite the research question being focused on the processing of noncontrast CT images, we did not exclude the MeSH term “angiography” from the query to avoid omissions of comparative analysis or studies with mixed target datasets. Suitable MeSH terms and keywords were identified using PubMed tools [17] and the Yale MeSH Analyzer [18]. Additionally, the most repeated words (except function words) were identified for a subset of five studies [19,20,21,22,23] by full-text automatic semantic analysis using an in-house developed Python script. To avoid extra bias, we did not include the MeSH term “sensitivity and specificity” in the query. The search strategies and queries are shown in Appendix A.

### 2.2. Data Collection and Analysis

We exported all articles identified in the database searches into the Mendeley Reference Manager [24], where duplicates were removed. Narrative review papers, commentaries, and letters to the editor were excluded. Two reviewers independently screened the titles and abstracts of all articles for eligibility. We emailed the authors if we were unable to retrieve the full paper, requesting a copy of the full publication. Authors were re-emailed after two weeks in the case of nonresponse, and if no contact had been made after three weeks, the study was excluded. The same pathway was used if the relevant data were not available in the published report. All full-text articles were independently and in duplicate screened for suitability, and reasons for exclusion were recorded. Any discrepancies in opinion between reviewers were discussed, the third reviewer was consulted in the case of disagreement.

Two reviewers independently identified and extracted the following data from each publication: study authors, country of origin, study design, sample size (including training and validation sets), dataset structure, test details and technical parameters (both for index test and reference test), and outcome measures. When possible, we extracted 2 × 2 contingency tables or summary statistics, from which they could be computed. If a study stratified the results by the aneurysm size, we divided the data into subgroups. If the number of included studies was small or of high heterogeneity, we summarized the key study-level information and synthesized the findings narratively, focusing on AI sensitivity and specificity. We also extracted Dice similarity coefficient (DSC) values, because this metric allows estimation of segmentation quality, essential for single-case studies. If there was no information about this metric, we calculated it from the presented images of the AI-segmented mask and GT mask according to the formula below [25]:(1)$$DSC=\frac{2|X\cap Y|}{\left|X|\cup |Y\right|},$$
where *X* represented the coordinates of the AI-segmented mask pixels, and *Y* represented the coordinates of the GT mask pixels. For this task, we exported the presented images of segmentation and GT (or original image) in JPEG format. If there were no expert markups, our medical expert (a certified radiologist with experience of 3 years) segmented it manually with a stylus using the Procreate 5.2.6 application [26] on iPad Pro 11. Then both pictures were binarized: the area was white inside the mask (values equal to 1) and black outside (values equal to 0), and they were aligned and analyzed automatically with an in-house developed script (Figure 1), prepared with R 4.1.2 [27].

We assessed the risk of bias and applicability concerns independently and in duplicate. Any disagreements were resolved through discussion. We did not use the QUADAS-2 domain list, as it was shown neither to accommodate the niche terminology encountered nor to signal the sources of bias found within AI studies [28]. Instead, we developed and used AI-specialized domain questions based on the traditional QUADAS-2 [29] (detailed information is presented in Appendix B).

## 3. Results

In total, we identified and imported 730 search results from PubMed into a Mendeley library. No additional relevant information was found in the grey literature sources. After title and abstract screening, 695 records were removed, including duplicates. Of the 35 studies selected for full-text assessment, we included eight studies in this review. Refer to Figure 2 for the PRISMA flow diagram of the search and inclusion results [12]. Exclusions were mainly due to ineligible study design (23 studies), ineligible study outcomes (three studies), or the absence of results (one study).

### 3.1. Description of Included Studies

We included eight studies (three journal articles and five conference papers) with a total of 355 cases, of which 273 (77%) had the diagnosis of AAA (Table 1). The studies were widely geographically distributed: three studies from the USA and one study each from Croatia, Greece, Japan, Iran, and Malaysia. Only three studies (37.5%) reported the data origin sources. These three studies presented algorithms based on neural network (NN) approach [23,30,31], and data augmentation was used in two of these works [23,31]. Other studies proposed different non-NN models and did not report any information regarding the data source or the expertise of the data tagging specialists. The examined outcomes were variable. Four articles did not present any quantitative metrics of AI accuracy [32,33,34,35]. Two articles did not present suitable images for DSC calculation. An example of the processing for the cases with the highest and the lowest DSC is presented in Figure 3.

### 3.2. Dataset Characteristics

The datasets of the included studies can be grouped in several ways. More than half of the studies (62.5%) were a “single-study” (or contained a single noncontrast series). Only one study used a full noncontrast CT: a single AAA-positive case consisting of 145 slices. There were four studies that used mixed datasets (noncontrast and contrast-enhanced CT images) [23,30,32,33]. Two studies [23,30] used representative sets, consisting of 321 (with 232 slices per study on average) and 10 (each case consisted of 160 slices) studies with pathology rates of 77% and 20%, respectively. Two other studies [32,33] used single cases consisting of 170 and 40 slices, respectively. Three studies [31,34,35] did not report the contrast usage in the CT examination. The overall data contained 21 AAA-positive cases with variable slice numbers: the mean number of slices in each case was 186 [31] and was of a single-case for two others.

### 3.3. Findings

The mean values for the relevant outcomes were as follows: 95% (95% CI 100–87%; three studies) for the sensitivity, 96.6% (95% CI 100–75.7%; two studies) for the specificity, 95.2% (95% CI 100–54.5%; two studies) for the accuracy, and 0.91 (95% CI 0.97–0.84; two studies) for the DSC. Only two studies simultaneously reported the accuracy, DSC, and sensitivity. Four studies did not report any quantitative diagnostic metrics. It was possible to calculate the DSC for six (75%) studies, and for two of them reporting the DSC, our calculation corresponded to the author-provided values. The data on the number of TP, FP, TN, and FN cases were provided only in one study [23] (Table 2). We tried to contact other authors to clarify the missing values, but unfortunately, the necessary data were not provided (either there was no answer or the authors had no information). There were also several design drawbacks, e.g., only two publications included nonpathological cases in the testing dataset. This made the quantitative estimation of the sensitivity, specificity, and accuracy impossible. One study [36] computed the sensitivity and positive predictive value by dividing a single case into two parts. The whole set consisted of 145 noncontrast CT scans of which 111 had AAA. For training and improving accuracy, the authors used 30 and 9 manually segmented noncontrast CT scans, respectively. No other dataset was used for algorithm validation. We suppose that this approach cannot be completely satisfactory, as the near-slice connection of the ROI (aortic lumen) introduced bias to the estimates.

### 3.4. Methodological Quality of Included Studies

The risk of bias due to the imbalanced dataset usage was high in six (75%) and low in two (25%) studies (Figure 4); the main concern was associated with the dataset imbalance in terms of the pathology and demographic ratios. The risk of bias due to concerns regarding the AI algorithm implementation was unclear in one (12.5%) and low in seven (87.5%) studies. These results were due to the fact that non-NN algorithms were used in half of the studies (thus, some QUADAS questions were irrelevant). The risk of bias due to the ground truth labeling concerns was unclear in four (50%) and low in four (50%) studies. The main concern was associated with the low clarity of the human readers’ expertise. Finally, the risk of bias due to the use of heterogeneous data was unclear in two (25%) and low in six (75%) studies. The reason for the ambiguity was connected to the low detail of the data processing pathway. The weights for each study were assigned proportionally to the number of processed cases. Additional details about the risk of bias assessment are provided in Appendix B. Concerns about the applicability for all domains were low for all studies, because the authors clearly postulated the research question, and their data, index, and reference tests were prepared and performed according to the claimed task.

## 4. Discussion

This systematic review summarized the published data on the application of AI for the automatic detection of AAA on noncontrast CT images and included eight unique studies. The major findings from our review include the following:1.The AI sensitivity for AAA detection varied from 92 to 98.4% with a mean value of 95% (95% CI 100–87%; three studies);2.The AI specificity for AAA detection varied from 95 to 98.3% with a mean value of 96.6% (95% CI 100–75.7%; two studies);3.The AI accuracy for AAA detection varied from 92 to 98.4% with a mean value of 95.2% (95% CI 100–54.5%; two studies);4.The DSC for AAA segmentation varied from 0.93 to 0.99 with a mean value of 0.96 (95% CI 0.99–0.94; two studies).

Since it was possible to perform only one measurement for DSC calculation, we considered the obtained values as estimates of the mean for segmentation quality. However, we observed a discrepancy between our measurements and two provided DSC values [30,31]. For the first algorithm, the reported mean DSC value was 0.91 versus our calculated 0.96. For the second one, the reported value was 0.9 versus our 0.99. We assume that the authors may have presented the best-case scenario for their algorithms, which could differ significantly from their real-life performance. These reported estimates of segmentation accuracy could be inflated. Therefore, we encourage authors to include examples of failed or suboptimal segmentation in order to access real-world applicability of the algorithms.

The success of the application of AI for the automatic detection of AAA on CTA has been previously reported by many researchers and has been already systematically reviewed [37]. At the same time, less attention has been paid to the AI-based screening capabilities. Screening tasks are usually performed with restricted timing, without contrast enhancement, and involve big data analysis. Our purpose was to investigate whether AI was applicable for tasks of AAA detection on CT without contrast enhancement. The reported AI sensitivity (95%) for AAA detection in noncontrast CT was higher than the AAA incidental detection sensitivity by radiologists (65%) [7]. Thus, AI may have the potential for AAA opportunistic screening automatization to increase the early detection of this pathology. However, due to objective reasons, this paper was unable to conduct a complete meta-analysis of the AI diagnostic accuracy. Moreover, the methodological quality analysis revealed several significant shortcomings of the included studies, causing serious doubts about the plausibility and reproducibility of the obtained metrics. In our opinion, the lack of regulations and reporting standards may be the reasons for the AI metrics’ overestimation in the original studies. To this end, STARD-AI recommendations for diagnostic accuracy studies are currently being developed [38].

### 4.1. Limitations of the Review

Our study had several limitations. Despite our results demonstrating the high diagnostic accuracy of AI for the automatic detection of AAA detection on a noncontrast CT, there were some concerns on the applicability and safety of the reviewed models in a clinical setting. The main reasons for the concerns were the sampling bias and the hidden stratification. Only two studies (25%) included nonpathological cases in the testing datasets. Moreover, 62.5% of included studies used a single AAA-positive CT scan to validate their algorithm, which does not allow estimation of the accuracy, specificity, and sensitivity. Because of this, we believe that the reported values of the sensitivity and specificity may be artificially high and need to be reassessed using standardized protocol and a high-quality independent testing dataset. Only a few papers met the inclusion criteria. However, the number of studies is not as important as their methodological quality: even if there were more studies, the methodological flaws and inflated diagnostic accuracy values cause doubts of the feasibility of meta-analysis. This is a well-known problem of reviews of AI studies [39] that requires regulatory attention. Perhaps consideration should be given not only to the reporting standardization of papers on diagnostic accuracy (STARD-AI) but also to AI-specific analyses in systematic reviews of such papers.

### 4.2. Implications of the Results for Practice, Policy, and Future Research

Our study revealed a significant difference in the number of studies on the detection of AAA from CT images with (over 500 studies) and without (eight studies) contrast enhancement. Nevertheless, despite its objective technical complexity, we consider the task of AAA detection from noncontrast CT scans just as clinically important, and we are looking forward to obtaining the results of the pilot project on AAA opportunistic screening [40] in the Moscow Experiment on Computer Vision in Radiology [41].

## 5. Conclusions

The uncertainty resulting from the high or unclear risk of bias associated with the heterogeneous parameters of the datasets (pathology ratio, studies per dataset, and slices per CT scan) limit our ability to confidently draw conclusions based on our results. Moreover, all eight studies included in the analysis evaluated automated AAA detection and segmentation on noncontrast CT using different accuracy metrics. To pool the accuracy values, we developed an original approach to approximate the DSCs from the imaging data included in the studies. According to our estimates, the algorithms in the included studies demonstrated high segmentation quality (DSC 0.96 ± 0.02). However, our results overestimated the DSC values provided by the authors (0.99 versus 0.9, and 0.96 versus 0.91), indicating a trend towards showcasing only the best examples of the algorithm’s performance, and the limited applicability of this approach. During the literature search, we observed an evident tendency in the published studies towards the use of contrast-enhanced scans for analysis (over 500 studies with CTA versus eight with noncontrast CT). Despite the higher task complexity of AAA detection and segmentation on noncontrast scans, it remains a promising meeting point for opportunistic screening prerequisites and practical computer vision implementation. Further studies are required, focused on balanced datasets with noncontrast CT scans and the utilization of reporting standards for satisfactory results’ reproducibility.

## Figures and Tables

**Figure 1 diagnostics-12-03197-f001:**
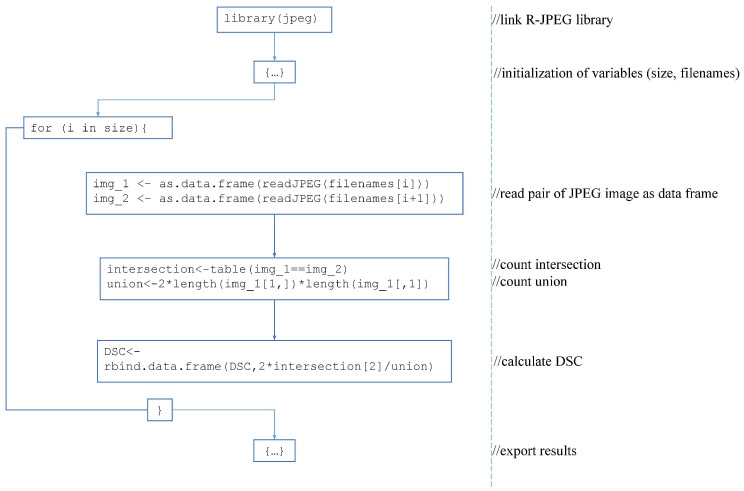
Scheme of R script for DSC calculation with comments.

**Figure 2 diagnostics-12-03197-f002:**
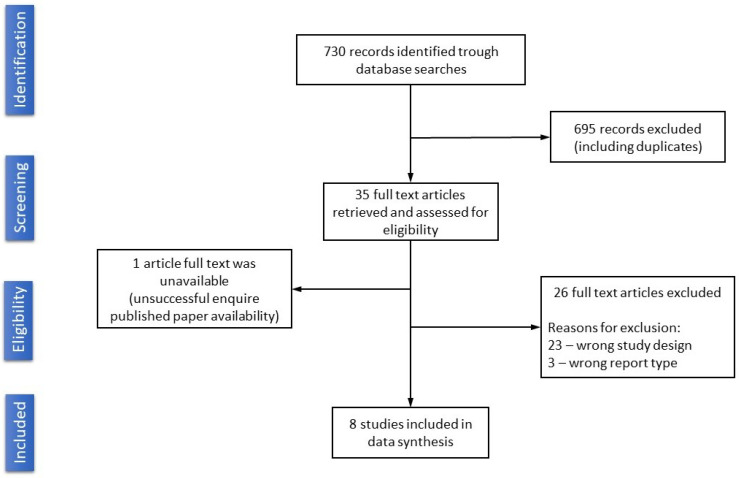
Flow diagram.

**Figure 3 diagnostics-12-03197-f003:**
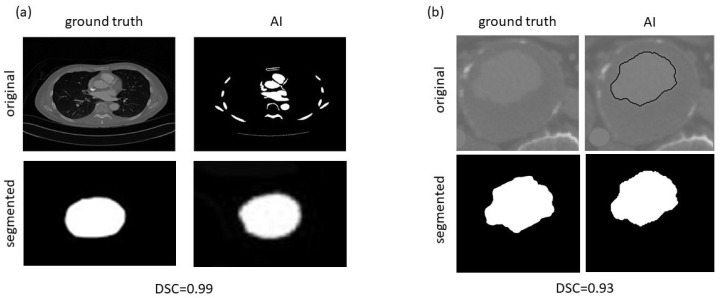
Image extraction for cases with the highest (**a**) and the lowest (**b**) DSC.

**Figure 4 diagnostics-12-03197-f004:**
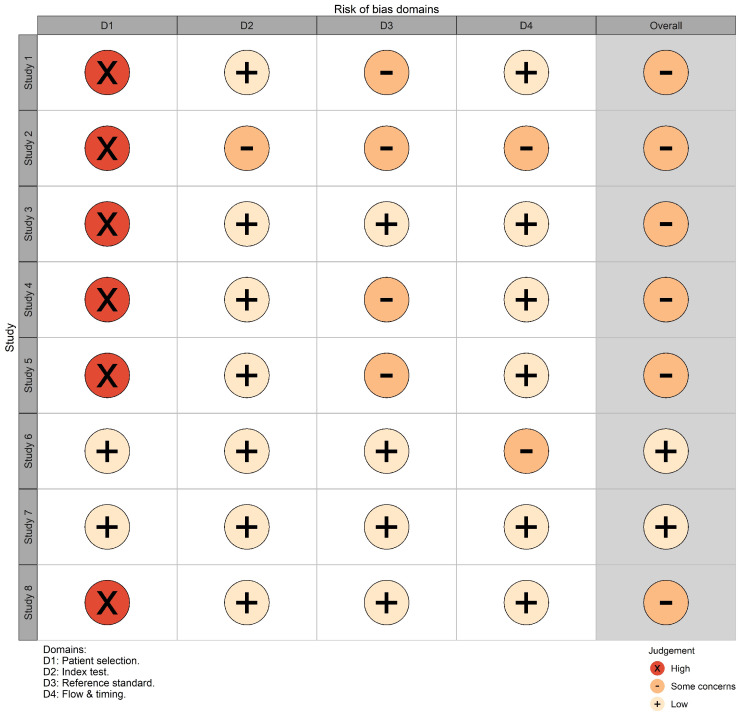
Risk of bias domains.

**Table 1 diagnostics-12-03197-t001:** Key characteristics of the studies ^1^.

№	1st Author (Year)	Study/Data Origin (Country)	Objectives	Type of Data Processing	Key Characteristics of Datasets	Relevant Outcomes	Calculated DSC
1	Almuntashri A. (2012) [32]	USA/-	AAA segmentation	Digital image processing algorithms	Two studies (one noncontrast case), 100% pathology rate, mixed	-	0.94
2	Fujiwara J. F. (2021) [36]	Japan/-	AAA detection and measurement	NN (not specified)	A single study, 100% pathology rate, noncontrast CT	Se 94.6%	-
3	Habijan M. (2020) [31]	Croatia/Belgium	AAA segmentation	NN (fourfold cross validation)	19 studies, 100 % pathology rate, CT type n/s	DSC 0.91 ± 0.16	0.96
4	Hosseini B. (2010) [33]	Malaysia/-	AAA detection	Non-NN (logical algorithm)	Two studies (one noncontrast case), 100% pathology rate, mixed	-	0.99
5	Kossioris G. T. (2008) [34]	Greece/-	AAA segmentation	Non-NN (level set method)	A single study, 100% pathology rate, CT type n/s	-	0.93
6	Lu J.-T. (2019) [30]	USA/USA	AAA detection	NN (fivefold cross validation)	321 studies, 77% pathology rate, mixed	Ac 92.0 %; Se 92.0%; Sp 95.0%; DSC 0.90 ± 0.05	0.99
7	Mohhamadi S. (2019) [23]	Iran/Iran	AAA segmentation and classification	Hough’s algorithm and NN (fivefold cross validation)	10 studies, 20% pathology rate, mixed	Ac 98.4%; Se 98.4%; Sp 98.3%	-
8	Schei T. R. (2003) [35]	USA/-	AAA detection	Non-NN (computer algorithm)	A single study, 100% pathology rate, CT type n/s	-	0.97

^1^ Note: used abbreviations: Se—sensitivity; Sp—specificity; Ac—accuracy.

**Table 2 diagnostics-12-03197-t002:** Data presence for confusion matrix arrangement ^1^.

№	Study First Author (Year)	Test Set Size (Images)	TP	FP	TN	FN
1	Almuntashri A. (2012) [32]	40	no information			
2	Fujiwara J. F. (2021) [36]	9				
3	Habijan M. (2020) [31]	not stated				
4	Hosseini B. (2010) [33]	170				
5	Kossioris G. T. (2008) [34]	1				
6	Lu J.-T. (2019) [30]	57				
7	Mohhamadi S. (2019) [23]	1448	357	11	1080	5
8	Schei T. R. (2003) [35]	1	no information			

^1^ Note: TP—true positive, FP—false positive, TN—true negative, FN—false negative (responses).

## Data Availability

Data are contained within the article, additional details are available upon request.

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
