# Peer review of "Diagnostic Accuracy of AI for Opportunistic Screening of Abdominal Aortic Aneurysm in CT: A Systematic Review and Narrative Synthesis"

_diagnostics, 2022, doi:10.3390/diagnostics12123197_

Round 1

Reviewer 1 Report

Topic (applicability of artificial intelligence (AI) in detecting AAA) is of interest and significance not only to the research community but also to medical practitioners, health-care services/providers, and general public. The manuscript is, in general, clearly and well written. However, the methods used require further clarification/explanation. Detailed comments and questions are below.

1)     Line 21: “AAA screening”: To the best of my knowledge, there are no AAA screening managed/used by public/private health systems. AAA is typically diagnosed by coincidence/opportunistically (as a “byproduct” for acquiring images for other purposes). If the Authors have information about the AAA screening programmes, the references need to be provided.

2)     Line 64-67: The Authors state that manual segmentation was used as a ground-truth. Does it mean that the Authors determined the aorta/AAA diameter from the segmented images, treated diameters greater than 30 mm as an indication of AAA (diagnosed AAA), and compared their results with the literature in which AI algorithms for image analysis were applied to detect AAA?

a.     How exactly the aorta/AAA diameter was determined from the images? Was any of the procedures clinically used (as described in Practice Guides {Wanhainen, 2019 #6757}) in AAA diagnosis  applied?

b.     What steps were undertaken to make the segmentation consistent between different analysts and mitigate the analyst’s bias always present in the manual segmentation?

c.      Was any image pre-processing conducted before the segmentation?

d.     How many analysts conducted the segmentation for a given patient?

e.     What was resolution of the images used in the study?

f.      Examples of the aorta/AAA segmentation and diameter measurement need to be included in the manuscript. Figure 3 shows only one section (transverse slice) through the abdominal area.

g.     How the segmentation accuracy was estimated/measured?

h.     What software tools/code were/was used for image segmentation?

3)     Line 170-173: “As a way out, one study [34] computed sensitivity and positive predictive value by dividing a single case into two parts”.

               Please explain how the dividing a “single case” into two parts was done?

4)     Providing the source data (i.e. segmented images) as supplementary data or Data in Brief article would strengthen the study and benefit the readers.

Author Response

Response to Reviewer 1 Comments

Point 1: Line 21: “AAA screening”: To the best of my knowledge, there are no AAA screening managed/used by public/private health systems. AAA is typically diagnosed by coincidence/opportunistically (as a “byproduct” for acquiring images for other purposes). If the Authors have information about the AAA screening programmes, the references need to be provided.

Response 1: Thank you for your comment!

Following your recommendation, we have added two more references [4, 5] to the existing one on line 22 that summarize the data on screening programs.

We referenced the «2014 ESC Guidelines on the diagnosis and treatment of aortic diseases» in our manuscript. This document contains an extensive section on AAA screening.

Population-wide AAA screening programs are currently proposed in several countries [1] with mixed results due to difficulties over implementation [2]. Several countries have not implemented such a program, despite national guidelines in favor of AAA screening. We are also aware of a number of abdominal aortic aneurysm screening programs conducted in different countries and systematic review of guideline documents [3-5].

[1] Stather PW, Dattani N, Bown MJ, Earnshaw JJ, Lees TA. International variations in AAA screening. Eur J Vasc Endovasc Surg 2013;45:231 -234.

[2] Shreibati JB, Baker LC, Hlatky MA, Mell MW. Impact of the Screening Abdominal Aortic Aneurysms Very Efficiently (SAAAVE) Act on abdominal ultrasonography use among Medicare beneficiaries. Arch Intern Med 2012;172:1456 -1462.

[3] Johansson, M., Zahl, P. H., Siersma, V., Jørgensen, K. J., Marklund, B., & Brodersen, J. (2018). Benefits and harms of screening men for abdominal aortic aneurysm in Sweden: a registry-based cohort study. The Lancet, 391(10138), 2441-2447.

[4] Mussa, F. F. (2015). Screening for abdominal aortic aneurysm. Journal of Vascular Surgery, 62(3), 774-778.

[5] Ferket, B. S., Grootenboer, N., Colkesen, E. B., Visser, J. J., van Sambeek, M. R., Spronk, S., ... & Hunink, M. M. (2012). Systematic review of guidelines on abdominal aortic aneurysm screening. Journal of vascular surgery, 55(5), 1296-1304.

Point 2: Line 64-67: The Authors state that manual segmentation was used as a ground-truth. Does it mean that the Authors determined the aorta/AAA diameter from the segmented images, treated diameters greater than 30 mm as an indication of AAA (diagnosed AAA), and compared their results with the literature in which AI algorithms for image analysis were applied to detect AAA?

  1. How exactly the aorta/AAA diameter was determined from the images? Was any of the procedures clinically used (as described in Practice Guides {Wanhainen, 2019 #6757}) in AAA diagnosis applied?
  2. What steps were undertaken to make the segmentation consistent between different analysts and mitigate the analyst’s bias always present in the manual segmentation?
  3. Was any image pre-processing conducted before the segmentation?
  4. How many analysts conducted the segmentation for a given patient?
  5. What was resolution of the images used in the study?
  6. Examples of the aorta/AAA segmentation and diameter measurement need to be included in the manuscript. Figure 3 shows only one section (transverse slice) through the abdominal area.
  7. How the segmentation accuracy was estimated/measured?
  8. What software tools/code were/was used for image segmentation?

Response 2: Thank you for your comment!

We are unable to give a detailed explanation of the items you specified in points a-h.

It was not possible to perform a meta-analysis of diagnostic accuracy metrics (i.e., independently calculate Se, Sp, DSC, etc.) due to the lack meta-data access (i.e., images labeled by experts and AI algorithms) in the analyzed articles. The only metric available for independent evaluation was DSC (Dice similarity coefficient) for pairs of images presented by the authors in the papers to illustrate their results. These images were extracted in JPEG format (which is not applicable for medical diagnostics) and processed using the developed method, detailed on lines 106-112.

Point 3: Line 170-173: “As a way out, one study [34] computed sensitivity and positive predictive value by dividing a single case into two parts”. Please explain how the dividing a “single case” into two parts was done?

Response 3: Thank you for your comment!

We added missed details to the lines 165-168.

Point 4: Providing the source data (i.e. segmented images) as supplementary data or Data in Brief article would strengthen the study and benefit the readers

Response 4: Thank you for your comment!

We decided to make this data available upon request, as we would like to collaborate with colleagues working with similar tasks. Our supplementary data contains the set of extracted images for DSC calculation and two designed scripts: one for article processing and other for DSC calculation.

Reviewer 2 Report

Table 1: please, start each line of the “Key characteristics of datasets” column with the capital letter, so to differentiate each previous line with each next line.

Discussion: “DSC for AAA segmentation varied from 0.93 to 1 with the mean value of 0.96 0.91 200 (95% CI 0.97 – 0.84; 2 studies)”: Some punctuation is missing, or something has been repeated twice.

Author Response

Response to Reviewer 2 Comments

Point 1: Table 1: please, start each line of the “Key characteristics of datasets” column with the capital letter, so to differentiate each previous line with each next line.

 Response 1: Thank you for your comment!

We have corrected this mistake.

Point 2: Discussion: “DSC for AAA segmentation varied from 0.93 to 1 with the mean value of 0.96 0.91 200 (95% CI 0.97 – 0.84; 2 studies)”: Some punctuation is missing, or something has been repeated twice.

Response 2: Thank you for your comment!

We have corrected this mistake.
